# Coccidioidomycosis: Changing Concepts and Knowledge Gaps

**DOI:** 10.3390/jof6040354

**Published:** 2020-12-10

**Authors:** Neil M. Ampel

**Affiliations:** Department of Infectious Diseases, Medicine and Immunobiology University of Arizona, 1501 North Campbell Avenue, Tucson, AZ 85724, USA; nampel@arizona.edu

**Keywords:** coccidioidomycosis, *Coccidioides*, antifungals, vaccines, ecology

## Abstract

Although first described more than 120 years ago, much remains unknown about coccidioidomycosis. In this review, new information that has led to changing concepts will be reviewed and remaining gaps in our knowledge will be discussed. In particular, new ideas regarding ecology and epidemiology, problems and promises of diagnosis, controversies over management, and the possibility of a vaccine will be covered.

## 1. Introduction

As stated more than two decades ago, coccidioidomycosis is a regional disease of national importance [1]. Recent studies attest to the continued medical [2] and economic impact [3] of this mycosis in the United States as well as in the rest of the western hemisphere [4,5]. In this review, particular aspects of coccidioidomycosis will be discussed that emphasize changing ideas and critical knowledge gaps. These will include new concepts regarding ecology, changing epidemiology, and possible geographic spread; problems and challenges with diagnosis; issues with treatment; and the possibility of a protective vaccine. While reference to older works will be noted when appropriate, the purpose here will be to cite more recent publications.

## 2. Defining the Ecology and Changing Epidemiology of Coccidioidomycosis

The distribution of coccidioidomycosis in nature was inferred in the past using three sources: epidemiology of proven cases, population-based skin test surveys, and ecological studies [6]. From these data, a model of coccidioidal infection was derived in which it was presumed that the fungus inhabits a soil niche and, either through direct exposure or due to air-borne spread, susceptible individuals acquire infection, usually through a respiratory route. While this model appears to be generally true, it is incomplete. With advances in molecular genomics, our understanding of the ecology and epidemiology of coccidioidomycosis is now changing and expanding. This began when it was established that *Coccidioides* consisted of two distinct species, *C. immitis* and *C. posadasii* [7]. Since then, the construction of phylogenetic trees has demonstrated that these species have distinct but overlapping geographic ranges [8,9] and have recently hybridized and exchanged genes [10]. Molecular genomics has now become refined enough to distinguish where a coccidioidal infection was geographically acquired [11].

However, the precise environments occupied by *Coccidioides* are still not adequately described. While it is clear that the fungus has a predilection for certain soils [12], it has been notoriously difficult to isolate using traditional culture methods [9] and its particular environmental niche remains unclear. Recently, the idea that small, soil-dwelling animals might serve as the primary environmental reservoir for *Coccidioides* [13], rather than the soil itself, has been revived. Taylor and Barker [14] have proposed that *Coccidioides* is not a saprophyte whose life cycle is a dead-end within the infected mammalian host, but rather an endozoan, able to live in the granulomata of infected mammals and, upon death of the host, use local nutrients released in the surrounding soil for nourishment. This concept is supported by genomic analysis of *Coccidioides* demonstrating, when compared to other filamentous Ascomycetes, that there are expansions of protease families and that the numbers of plant cell-wall degrading enzymes are reduced compared to fungi known to decay plant matter [15]. This hypothesis has been strengthened by experimental data. DNA extracted from 465 soil samples from five Arizona locations was subjected to real-time qPCR. Of these, 105 samples (23%) were positive for *Coccidioides* and 95 of these were from animal burrows [16]. These results suggest that *Coccidioides* is not extant in the soil environment, but rather resides in focal pockets in association with small animals and their burrows. These results are supported by studies in the northern Baja region of Mexico that have used both molecular detection of *Coccidioides* in the soil and antibody detection of small rodents [17,18]. Moreover, this hypothesis fits with the concept that certain fungi, including members of the order Onygenales to which *Coccidioides* belongs, may be part of the normal human lung microbiome [19].

The conditions for air-borne spread and respiratory acquisition have also remained elusive. It has been estimated that a single arthroconidium is sufficient to cause infection in an animal model [20]. However, there is a clinical sense that exposure to higher inocula results in more severe respiratory disease [21] and that the risk of extrapulmonary dissemination may be increased with exposure to higher respiratory inocula [22]. The median time of living in the coccidioidal endemic area in Arizona at the time of serological conversion has been estimated to be 6.5 years and appears to increase linearly with time [23]. However, the risk of infection also appears to be stochastic and dependent on specific occurrences, such as dust storms and earthquakes [24]. These data imply that a variety of events, many currently undefined, may play important roles in predicting whether an individual acquires coccidioidal infection.

In the past, assessing the air-borne spread of *Coccidioides* has been difficult. However, genomics may be improving this. In one recent study, air was sampled and filtered from 21 sites in the Phoenix metropolitan area on a specific day after a dust storm as well as over a 45-day period during the fall. DNA was extracted from the filters and subjected to nested qPCR [25]. Although prior evidence has suggested that dust storms are associated with outbreaks of coccidioidomycosis [26], there was actually a decrease in positive filter samples from sites on the days immediately after the dust storm compared to before. Moreover, there were both temporal and geographic differences in filter positivity during the 45-day sample period. Clearly, more such studies are needed and a linkage to human cases will be required to flesh out associations. However, the ability to consistently detect *Coccidioides* in the air is a critical first step.

Finally, in recent years, environmental sites containing *Coccidioides* that are well outside of the traditionally recognized coccidioidal endemic zones have been associated with outbreaks of coccidioidomycosis. These include an ancient Amerindian location in northeast Utah [27] and an area of eastern Washington state [28]. Moreover, the areas where *Coccidioides* might reside appears to be much larger than the sites already identified [29]. Because the organisms at these sites have likely been there for millennia, one must conclude that these outbreaks are associated with some recent alteration, such as climate change or human population expansion. Recently, Gorris and her colleagues have suggested [30], using a county-level database of cases and climatic and environmental information, that the endemic region of coccidioidomycosis is likely to extend beyond currently recognized boundaries and that it will increase within the known endemic regions as the climate changes.

## 3. Issues with Diagnosis

While coccidioidomycosis may be diagnosed in a variety of ways, serology has by far remained the most common method used. Because most clinical laboratories have the appropriate platform and because the results are generally available within 24 h, enzyme immunoassays (EIAs) are most frequently employed. However, there is no established standard by which the several available commercial EIAs can be compared and all use proprietary antigens that are not described [31]. This had led to a certain amount of uncertainty with regard to their meaning and interpretation in certain clinical situations.

EIA coccidioidal antibody testing is particularly useful among patients who present with a typical clinical syndrome of coccidioidomycosis within the endemic area [32]. In such instances, when both the IgM and IgM assays are positive, it is highly likely that the patient has primary pulmonary coccidioidomycosis. However, these tests are less useful in other situations, such as screening immunocompromised patients [33]. This was recently emphasized in a retrospective review of rheumatology patients in the endemic area who were routinely screened for coccidioidal antibody [31]. In that study, the finding of a positive coccidioidal serology in the absence of symptoms did not predict clinical illness.

Recently, McHardy and colleagues reviewed their experience at the University of California at Davis with the coccidioidal complement fixation (CF) test in the age of antifungal therapy [34]. From their large database, they found that titers generally reflected clinical illness, with the lowest ones found in those with uncomplicated pulmonary disease and higher levels with disseminated extrathoracic illness. While they did not find a specific titer predictive of disseminated disease, a titer ≥ 1:32 suggested a complicated clinical course. They also noted that the change in CF titers was slow, with a decline by one dilution every 90 days for those with uncomplicated pulmonary disease and longer for other patients. These data suggest that the test should be ordered at appropriately long intervals to be used as a measure of improvement.

Delays in diagnosis of coccidioidomycosis have been shown to be costly [35] and more rapid diagnostic tests are needed. Recently, a lateral flow antibody assay (LFA) that could allow bedside performance with results in under one hour has been developed [36]. Unfortunately, a recent assessment suggests that it may have an unacceptably low sensitivity when compared to EIA [37]. Direct assays of *Coccidioides* are also available. The measurement of β-1,3-d-glucan (BDG) is a non-specific fungal assay that is now widely available in clinical laboratories. Unfortunately, in one study, it had a sensitivity of only 19% at a cut-off level of ≥80 pg/mL among patients with acute pulmonary coccidioidomycosis [38]. More promising is PCR assessment. These assays have been shown to be rapid, specific, and recently capable of being performed in clinical microbiology laboratories [39,40]. Although the overall sensitivity, particularly compared to serology, is not known, these tests could become very useful if they were to become more generally available.

There has been renewed interest in assessing the cellular immune response to coccidioidomycosis as both a diagnostic and a prognostic tool. In a study of a reformulated spherule-based reagent [41], 52 of 53 subjects (98%) with acute pulmonary coccidioidomycosis demonstrated ≥5 mm of induration 48 h after intradermal injection. Based on this, the test was approved in 2014 for use in the United States. A subsequent study of 36,789 inmates in a prison system demonstrated an 8.6% positivity rate that correlated with residence in the endemic region [42]. The results were used to restrict inmates with negative tests from incarceration at two prisons with the highest coccidioidomycosis rates. The use of the test for prognostic purposes is less clear. Two recent studies of patients followed for various types of coccidioidomycosis have suggested that expression of delayed type hypersensitivity reactions following skin testing may not always correlate with clinical expression of disease or be prognostically helpful [43,44].

Intradermal skin testing has several drawbacks. It requires skill and training both in the placement and in the interpretation of the result and it requires an additional visit 48 h after placement to read the response. In addition, there is a predictable incidence of adverse events, nearly 5% among those in the prison study [42]. Perhaps because of these issues, the test has not been extensively adopted for use in the United States since its introduction [45]. There has been interest in developing a whole blood assay measuring cytokine release after ex vivo antigen exposure. This method has been shown to correlate with skin test dermal hypersensitivity [46] and has been studied in several experimental models [47,48,49]. However, at this time, no commercial assay is available.

Several tests are on the horizon. Recently, Peng and colleagues [50] reported using recombinant CST1, a fungal chitinase that is the antigen for the IgG antibody response [51], to perform quantitative coccidioidal titers using an enzyme-linked system. They were able to identify a 200-amino acid segment of the gene product that specifically bound antibody. The assay correlated well with the standard CF assay but with the potential advantage that the test could be performed by many clinical laboratories in much less time than the current method.

*Coccidioides* produces a variety of volatile organic compounds (VOC) that can be detected using solid phase microextraction (SPME) and comprehensive two-dimensional gas chromatography time-of-flight mass spectrometry (GC × GC-TOFMS). The number of these metabolites was found to be increased in a wild-type strain of *C. posadasii* compared to an attenuated mutant strain [52]. Detecting such gas-phase metabolites could be used as a potentially rapid diagnostic tool through breath analysis. Preliminary data have suggested the feasibility of this in an experimental model studying multiple strains of *C. posadasii* and *C. immitis* [53].

Finally, using a targeted liquid chromatography-tandem mass spectrometry-based (LC-MS/MS) metabolic profiling approach, 207 plasma and 231 urinary coccidioidal metabolites were reliably detected in high abundance in samples from 48 patients with clinical coccidioidomycosis compared to 99 individuals without that diagnosis [54]. Statistical modeling allowed the identification of three significantly altered plasma metabolites and nine urine metabolites in patients with coccidioidomycosis. Constructing receiver–operator curves demonstrated that the plasma metabolites had a sensitivity of >94% and a specificity of >97%. Because LC-MS/MS is now available in many clinical laboratories, this too could become a rapid assay for the diagnosis of coccidioidomycosis.

## 4. Treatment: When and What

Prior to the development of antifungal drugs in the 1950s, it was demonstrated that fully 60% of individuals who acquired coccidioidal infection did so without symptoms and the vast majority of those with symptoms resolved their clinical disease without sequelae and with apparent long-lived immunity [55]. The relative benignity of pulmonary compared to extrathoracic disseminated coccidioidomycosis in the age before antifungals was recently demonstrated in a retrospective review of 531 patients followed until 1966 as part of the VA-Armed Forces Cooperative Study [56]. In that study, the all-cause mortality of pulmonary coccidioidomycosis was 5.4% over 30 years, significantly lower than 30% in those with non-meningeal disseminated disease and 96% in those with meningeal coccidioidomycosis.

With the development of oral triazole antifungal therapy in the 1990s, treatment of symptomatic primary pulmonary coccidioidomycosis became a common practice, despite the lack of controlled trials. Two retrospective cohort studies have suggested that there may be no clear benefit from antifungal therapy for such patients. In the first [57], 43 patients with primary pulmonary coccidioidomycosis were followed for a median 286 days. Among 16 who received triazole antifungal therapy, at the discretion of the treating providers, two developed extrathoracic dissemination after treatment was discontinued, while all 36 who received no therapy had an uncomplicated outcome. Moreover, the rate of improvement was not different between the two groups. In the second study [58], 27 patients with primary pulmonary coccidioidomycosis were followed for 24 weeks. Twenty patients, again at the discretion of the providers, were prescribed antifungal therapy. As in the previous study, median times to improvement between the two groups were not different and one patient developed extrathoracic dissemination; that patient was in the treatment group. Because both of these studies were not controlled with regard to therapy, they were potentially biased. In an attempt to definitively answer whether early antifungal therapy leads to improved outcomes for those with primary pulmonary coccidioidomycosis, the National Institute of Allergy and Infectious Diseases (NIAID) sponsored a study that provided 42 days of fluconazole or placebo for patients living in the coccidioidal endemic region with presumed coccidioidal pneumonia (Clinical trials.gov identifier: NCT02663674). Unfortunately, this trial ended early due to lack of enrollment. For now, it seems that patients without underlying conditions and without evidence of severe infection may not require antifungal therapy for uncomplicated primary pulmonary coccidioidomycosis.

A second important therapy question is what antifungal agent is best to use for coccidioidomycosis. There has be a consistent deference toward using fluconazole in this situation, both because this was the triazole antifungal initially studied in coccidioidomycosis and because it was perceived to have the most benign toxicity profile. However, in fact, there are data to challenge this view. First, in early non-comparative studies of chronic coccidioidomycosis, relapses were less frequent in those who received itraconazole compared to fluconazole [59,60]. In the only randomized, double-blind comparative trial of fluconazole and itraconazole for chronic non-meningeal coccidioidomycosis, the response rate to itraconazole was significantly higher than fluconazole in those with skeletal infections [61]. Moreover, when patients fail fluconazole, a switch to another triazole is frequently effective [62]. The preferred therapy for two related fungal infections, histoplasmosis and blastomycosis, is not fluconazole but rather itraconazole. Finally, two in vitro susceptibility studies have demonstrated that, of all the available triazole antifungals, fluconazole has by far the highest minimum inhibitory concentrations (MICs) against *Coccidioides* [63,64] (Table 1). Unfortunately, there are no recent comparative trials of triazole antifungals for coccidioidomycosis. These are now urgently needed. In the meantime, a reevaluation of the deference toward the use of fluconazole for coccidioidomycosis should be reconsidered.

The triazole antifungals have distinct toxicities inherently related to their mechanism of action. All exert their antifungal effect by preventing the conversion of lanosterol to ergosterol by inhibiting 14-α-demethylase [65], a reaction dependent upon CYP51, which is also present in humans [66]. In addition, cross-inhibition of several human CYP-dependent enzymes, particularly CYP3A4, 2C9, and 2C19, is responsible for many of the clinical side effects and drug–drug interactions. The latter are of particular concern, as many pharmacological agents depend on these systems for their metabolism. Inhibition of these metabolic pathways by triazole antifungals can result in significant toxicity [67]. Particularly common is the increase in coagulation seen when these agents are combined with warfarin and the increased in serum levels and toxicity that occur when combined with the calcineurin inhibitors cyclosporine and tacrolimus. Although fluconazole has been considered the least toxic of the triazole antifungals, a long-term use has been shown to result in adverse effects in the majority of patients. Among the most common are xerosis, alopecia, and fatigue [68]. The triazole antifungals have also been associated with teratogenicity, an effect that appears to be directly related to inhibition of human CYP51. They should be avoided in women seeking to become pregnant and during the first trimester of pregnancy [69]. There are also unique toxicities. Voriconazole, which is trifluorinated, has been associated with periostitis due to elevated fluoride levels [70,71]. Recently, the delayed-release formulation of posaconazole has been associated with the syndrome of pseudoaldosteronism [72]. Patients have presented with severe hypertension and hypokalemia often associated with elevated drug levels that resolves with stopping or reducing the dosage of the medication. The mechanism appears related to the inhibition of 11β-hydroxysteroid activity [73].

## 5. Antifungals on the Horizon

Nikkomycin Z (Table 2) is a competitive inhibitor of chitin synthase because of its structural similarity to uridine diphosphate-*N*-acetylglucosamine, a monosaccharide building block of fungal chitin. The target is lacking in human hosts [74]. Hector and colleagues [75] demonstrated significant in vitro activity against both *Coccidioides* and *Blastomyces*, but less activity against yeast and none against *Aspergillus* species. Doses of 20 and 50 mg/kg were completely protective in a murine intranasal coccidioidal infection model. More importantly, treatment with the 50 mg/kg dose for six days eradicated nearly all of the fungus from the lungs of treated animals; a single colony of *Coccidioides* was recovered from one of eight mice infected and treated. This suggests that it is fungicidal and potentially curative. Li and Rinaldi subsequently demonstrated that combining nikkomycin Z with either fluconazole or itraconazole resulted in synergistic in vitro activity against a variety of fungal pathogens [76]. A pharmacological study in humans revealed that a single oral dose was well tolerated and resulted in no toxicity with therapeutic bioavailability and a half-life of 1–2 h [77]. Shubitz and colleagues completed treatment of nine canines with coccidioidomycosis in an open-label study at doses of either 250 mg or 500 mg daily for an average of three months. Most had been previously treated with fluconazole. Overall, seven of the dogs improved. No toxicity was observed and the twice-daily therapy resulted in acceptable pharmacokinetics [78]. Subsequently, it was shown that a dose of 80 mg/kg daily given in two doses for three weeks in mice resulted in optimal clearance of *Coccidioides* from the lungs with acceptable pharmacokinetics that could be achieved in humans [79]. Although nikkomycin Z has been granted qualified infectious diseases product (QIDP) designation by the Food and Drug Administration, at this time, it is not being studied further in humans until more is manufactured.

Tetrazoles are a new class of antifungals that have specific, selective activity against fungal but not mammalian 14-α-demethylase (CYP51) [74]. Because of this, these agents avoid many of the CYP-related toxicities associated with triazole antifungals detailed above. One of these, VT-1598 (Table 2), has been studied specifically against *Coccidioides*. Using an in vitro model, VT-1598 had MICs against clinical isolates of *Coccidioides* between 0.06 and 0.50 µg/mL, comparable to posaconazole and far below that of fluconazole [80]. In a murine model of coccidioidal meningitis, VT-1598 at 20 mg/kg, beginning two days after inoculation with either the Silveira strain of *C. posadasii* or a clinical isolate of *C. immitis*, resulted in protection from death that appeared to be superior to fluconazole at 25 mg/kg. Plasma levels remained in the inhibitory range two days after the last treatment [81]. Like nikkomycin Z, VT-1578 has QIDP designation and human studies are being planned.

Olorofim (Table 2) is a member of a new class of antifungals, the orotomides, that reversibly inhibits fungal pyrimidine biosynthesis through the enzyme dihydroorotate dehydrogenase (DHODH) with little inhibition of the human enzyme [82,83]. It has demonstrated in vitro activity against *Coccidioides* [80,82]. In a murine model of coccidioidal meningitis, doses of 10 and 20 mg/kg appeared to confer a survival advantage over fluconazole at 25 mg/kg, although there were still deaths [84]. An intravenous formulation using a β-hydroxypropyl-cyclodextrin vehicle has been studied in human volunteers with a half-life between 20 and 30 h. An oral formulation is also being studied [83]. Human studies for coccidioidomycosis are being planned.

## 6. Vaccines

As pointed out by Kirkland, immunization for coccidioidomycosis seems feasible since second infections appear to be extraordinarily rare and initial infection appears to confer life-long immunity [85]. Barnato and colleagues in 2001 proposed an economic case for the development of a vaccine using a decision model [86]. They estimated that adult vaccination in the coccidioidal endemic regions would result in a savings of $62,000 per quality adjusted life year compared to no vaccination and would result in 11 fewer deaths and save $3 million annually. Current data appear to be even more in favor of vaccination. In California during 2017, the estimated direct and indirect costs for 7466 patients was nearly $700 million [3]. In Arizona, the costs of managing delays in diagnoses were nearly $600,000 in just under a three-year period [35].

Because of this, there has been renewed interest in a coccidioidal vaccine since the publication in 1993 of the inconclusive results of a formaldehyde-killed thimerosal-preserved spherule vaccine [87,88]. For this renewed effort, the goals of vaccination would be multiple. A vaccine should provide protection against primary acute infection, prevent extrathoracic dissemination, and protection should persist for a prolonged if not life-long period and not wane during immunosuppression, and the vaccine should be safe and well-tolerated. 

Two approaches have been taken. In the first, a live mutant strain of *C. posadasii* has been created by deleting the *CPS1* gene, a fungal virulence factor. The resultant Δcsp1 strain demonstrates slower growth and smaller spherules than wild-type when grown under in vitro conditions. Moreover, the Δcsp1 strain was found to be avirulent in mice and, when used as a vaccine, resulted in protection from wild-type coccidioidal infection in susceptible mouse strains [89]. This has resulted in an NIAID sponsored study to develop the vaccine in a canine model. Results for this study are currently pending, but if successful, it could serve as a bridge for the development of a human vaccine.

The second direction is to develop a sub-unit plus adjuvant vaccine using antigens that have induced protection in mouse models of coccidioidomycosis. In this system, a construct using three *C. posadasii* antigens, Ag2-PRA, Cs-Ag, and Pmp-1, collectively called rCpa1, have been attached to a five-peptide sequence containing human T-cell epitopes that allows for binding to human HLA Class II molecules. This is then incorporated into glucan–chitin particles derived from *Rhodotorula mucilaginous*, which acts as the adjuvant. Preliminary studies in mice have shown protection against strains of both *C. immitis* and *C. posadasii* [90], stimulation of the Th1/Th17 pathway [91], and activation of the CARD9 dectin-1 and dectin-2 signaling pathways [92]. Finally, no significant toxicity has been seen in a human liver cell line (HepG2) (personal communication, C-Y Hung). We will have to await further results from both these endeavors, but it is likely that a vaccine for coccidioidomycosis will be developed for humans in the foreseeable future.

## Figures and Tables

**Table 1 jof-06-00354-t001:** Comparison of minimum inhibitory concentrations (µg/mL) of triazole antifungals.

Parameter	Fluconazole (*n* = 581)	Itraconazole (*n* = 486)	Voriconazole (*n* = 499)	Posaconazole (*n* = 377)
MIC_50_	8	0.250	0.125	0.125
MIC_90_	16	0.500	0.250	0.250
GM MIC	7.71	0.245	0.107	0.141

MIC_50_: minimum inhibitory concentration 50%. MIC_90_: minimum inhibitory concentration 90%. GM MIC: geometric mean minimum inhibitory concentration. Parentheses contain number of isolates tested. Adapted from Thompson et al. [62].

**Table 2 jof-06-00354-t002:** Antifungals on the horizon.

Agent	Mechanism of Action	Half-Life	Route of Delivery
Nikkomycin Z	competitive inhibition of chitin synthase	1–2 h	IV and oral
VT-1598	inhibition of 14-α-demethylase (CYP51)	24 h	oral
Olorofim	reversible inhibition pyrimidine biosynthesis	20–30 h	IV (with vehicle) and oral

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
