# Peer review of "Coccidioidomycosis: Changing Concepts and Knowledge Gaps"

_jof, 2020, doi:10.3390/jof6040354_

Round 1

Reviewer 1 Report

This is a well-written, excellent review of the status of research and conceptual thinking regarding the biology of Coccidioides and coccidioidomycosis. I have only a few minor comments.

Lines 4 and 6. Should “Neil M. Ampel and M.D.” be “Neil M. Ampel, M.D.”?

Line 31. Maybe mention evidence for interspecies hybridization: (https://genome.cshlp.org/content/20/7/938).

Paragraph beginning line 33. A recent review by Hamm et al. (https://doi.org/10.1371/journal.ppat.1008684) argues that both early and recent studies provide additional arguments that species of Coccidioides and other members of the Onygenales, notably Blastomyces parvus (aka Haplosporangium parvum), are commensal inhabitants of mammalian lung tissues that can become pathogenic when hosts are immune compromised (pathogenic potential almost certainly also influenced by inoculum potential in the case of Coccidioides). This commensalism hypothesis is interesting in the context of the information presented here in the paragraph that begins on line 169, which suggests the possibility that humans with primary pulmonary coccidioidomycosis recover independent of whether they are treated with antifungals.

Line 38. Should “had” be “has”?

Line 115. Is the repetition of “recently” needed here?

Line 181. Add “of” before “these”?

Line 288. Maybe reword “smaller spherules when grown under in vitro conditions than wild-type” to “smaller spherules than wild-type when grown under in vitro conditions.”

Author Response

Lines 4-6: the “and” has been deleted and replaced by a comma

Line 31: the line “and have recently hybridized and exchanged genes.” has been added with the reference 9a.

Line 33: a sentence has been added starting on the new line 52 indicating the presence of a possible fungal lung microbiome inhabited by Onygenales with the addition of the reference 17a. The author appreciates the linkage of commensalism to the fact that most individuals live with their coccidioidal infection,  but no further change was made in the manuscript regarding this.

Line 38: “the idea that small, soil-dwelling animals … has been revived.” The verb implies current action and seems more apt that “had” and has been retained. No changes were made.

Line 118: the first “recently” has been deleted.

Line 186: “of” has been added before “these”

Line 293: the phrase has been changed to “smaller spherules than wild-type when grown under in vitro conditions” as suggested.

Reviewer 2 Report

This is a very well written review of the current knowledge about Valley fever. These types of reviews are difficult to find, and so the paper should be very useful to those just entering the field.

Author Response

There were no specific issues to address from this reviewer.